# Assessing Performance and Fairness Metrics in Face Recognition - Bootstrap Methods

**Jean-Rémy Conti**
Télécom Paris, Idemia
jean-remy.conti@telecom-paris.fr

**Stéphan Clémençon**
Télécom Paris
stephan.clemencon@telecom-paris.fr

## Abstract

The ROC curve is the major tool for assessing not only the performance but also the fairness properties of a similarity scoring function in Face Recognition. In order to draw reliable conclusions based on empirical ROC analysis, evaluating accurately the uncertainty related to statistical versions of the ROC curves of interest is necessary. For this purpose, we explain in this paper that, because the True/False Acceptance Rates are of the form of U-statistics in the case of similarity scoring, the naive bootstrap approach is not valid here and that a dedicated recentering technique must be used instead. This is illustrated on real data of face images, when applied to several ROC-based metrics such as popular fairness metrics.

## 1 Face Recognition - Performance & Fairness

The deployment of Face Recognition (FR) systems brings with it a pressing demand for methodological tools to assess their trustworthiness. The reliability of FR systems concerns their estimated performance of course, but also their properties regarding fairness: ideally, the system should exhibit approximately the same performance, independently of the *sensitive* group (determined by *e.g.* gender, age group, race) to which it is applied. While, until now, the benchmark of FR systems is essentially reduced to an ad-hoc evaluation of the performance metrics (*i.e.* ROC analysis) on a face image dataset of reference, the purpose of this paper is to explain, and illustrate using real data how the bootstrap methodology can be used to quantify the uncertainty/variability of the performance metrics, as well as that of some popular fairness metrics. Hopefully, this paves the way for a more valuable and trustworthy comparative analysis of the merits and drawbacks of FR systems.

In FR, the usual objective is to learn an encoder function $f : \mathbb{R}^{h \times w \times c} \to \mathbb{R}^d$ that embeds the images in a way that brings same identities closer together. Each image is of size $(h, w)$, while $c$ corresponds to the color channel dimension. It is worth noting that a pre-processing detection step (finding a face within an image) is required to make all face images have the same size $(h, w)$. For an image $x \in \mathbb{R}^{h \times w \times c}$, its latent representation $f(x) \in \mathbb{R}^d$ is called the face embedding of $x$.

Since the advent of deep learning, the encoder $f$ is a deep Convolutional Neural Network (CNN) whose parameters are learned on a huge FR dataset, made of face images and identity labels. In brief, the training consists in taking all images $x_i^k$, labelled with identity $k$, computing their embeddings $f(x_i^k)$ and adjusting the parameters of $f$ so that those embeddings are as close as possible (for a given similarity measure) and as far as possible from the embeddings of identity $l \neq k$. The usual similarity measure is the cosine similarity which is defined as $s(x_i, x_j) := f(x_i)^\intercal f(x_j)/(||f(x_i)|| \cdot ||f(x_j)||)$ for two images $x_i, x_j$, with $|| \cdot ||$ standing for the usual Euclidean norm. In some early works [Schroff et al., 2015], the Euclidean metric $||f(x_i) - f(x_j)||$ was also used. In the rest of this document, we discard the notation $f$ for the encoder, and only use the similarity $s$ (which contains the encoder), as we are not interested in the encoder training.

2022 Trustworthy and Socially Responsible Machine Learning (TSRML 2022) co-located with NeurIPS 2022.

## 1.1 Performance Evaluation in Face Recognition

There are generally two FR use cases: *identification*, which consists in finding the specific identity of a probe face among several previously enrolled identities, and *verification* (which we focus on throughout this paper), which aims at deciding whether two face images correspond to the same identity or not. In practice, the evaluation of a trained FR model is achieved using an evaluation dataset, where all possible pairs $(x_i, x_j)$ of face images are considered. Then, an operating point $t \in [-1, 1]$ (threshold of acceptance) is chosen to classify the pair $(x_i, x_j)$ as *genuine* (same identity) if $s(x_i, x_j) > t$ and *impostor* (distinct identities) otherwise. In the following, we describe the statistical measures for evaluating a FR model, given an evaluation dataset.

Assuming that there are $K$ distinct identities, the evaluation dataset can be modeled by a random variable $(X, y) \in \mathbb{R}^{h \times w \times c} \times \{1, \ldots, K\}$. We denote by $\mathbf{P}$ the corresponding probability law. For $1 \leq k \leq K$, we assume that the identities are equiprobable i.e. $\mathbf{P}(y = k) = \frac{1}{K}$. $X$ is determined by its conditional distributions $X^k := (X | y = k) \sim \mathcal{I}_k$ and we consider that $X^k, X^l$ are independent if $k \neq l$.

Let $(X_1, y_1)$ and $(X_2, y_2)$ be two independent random variables with law $\mathbf{P}$. We distinguish between the False Negative Rate (FNR) and the True Negative Rate (TNR), respectively defined by:

$$F(t) = \mathbf{P}(s(X_1, X_2) \leq t \,|\, y_1 = y_2) \quad \text{and} \quad G(t) = \mathbf{P}(s(X_1, X_2) \leq t \,|\, y_1 \neq y_2).$$

With these notations, the ROC curve is defined as the graph of the mapping

$$\text{ROC} : \alpha \mapsto \text{ROC}(\alpha) = F \circ G^{-1}(1 - \alpha) \quad \text{with} \quad \alpha \in [0, 1].$$

Note that by $\text{ROC}(\alpha)$, one usually means $1 - F \circ G^{-1}(1 - \alpha)$ in machine learning and statistical literature but the FR community favors the DET curve $(1 - \text{ROC}(\alpha))$, which we will call ROC curve in the following.

In practice, those metrics are not computable since we only have a finite dataset. We denote by $n_k$ the number of face images of identity $k$, for $1 \leq k \leq K$, within the evaluation dataset. The images of identity $k$ are modeled by random variables $(X_i^k)_{1 \leq i \leq n_k}$, independent copies of $X^k$. The empirical approximations $F_N$ and $G_N$ of $F$ and $G$ are:

$$F_N(t) = \frac{1}{K} \sum_{k=1}^{K} \frac{1}{\binom{n_k}{2}} \sum_{1 \leq i < j \leq n_k} \mathbf{1}_{s(X_i^k, X_j^k) \leq t}$$

and

$$G_N(t) = \frac{1}{\binom{K}{2}} \sum_{1 \leq k < l \leq K} \frac{1}{n_k n_l} \sum_{\substack{1 \leq i \leq n_k \\ 1 \leq j \leq n_l}} \mathbf{1}_{s(X_i^k, X_j^l) \leq t}.$$

The empirical ROC curve is naturally:

$$\text{ROC}_N : \alpha \mapsto \text{ROC}_N(\alpha) = F_N \circ G_N^{-1}(1 - \alpha) \quad \text{with} \quad \alpha \in [0, 1]. \tag{1}$$

## 1.2 Fairness Metrics in Face Recognition

To be consistent with the FR community, we change our previous notations (only for addressing fairness metrics) and define the False Rejection Rate (FRR) and the False Acceptance Rate (FAR) respectively as $\text{FRR}(t) := F_N(t)$ and $\text{FAR}(t) := 1 - G_N(t)$. Both are error rates that should be minimized, one more than the other depending on the use case. With those notations, the empirical ROC curve is $\text{ROC}_N(\alpha) = \text{FRR}(t_\alpha)$ with $\text{FAR}(t_\alpha) = \alpha$.

In order to inspect fairness issues in FR, one should look at differentials in performance amongst several subgroups of the population. Those subgroups are distinguishable by a sensitive attribute (*e.g.* gender, race, age, ...). For a given discrete sensitive attribute that can take $A > 1$ different values, we enrich our previous model and consider a random variable $(X, y, a)$ where $a \in \mathcal{A} = \{0, 1, \ldots, A-1\}$. With a slight abuse of notations, we still denote by $\mathbf{P}$ the corresponding probability law and, for every fixed value $a$, we can further define

$$F^a(t) := \mathbf{P}(s(X_1, X_2) \leq t \,|\, y_1 = y_2, \, a_1 = a_2 = a)$$
$$G^a(t) := \mathbf{P}(s(X_1, X_2) \leq t \,|\, y_1 \neq y_2, \, a_1 = a_2 = a).$$

The empirical approximations of $F^a(t)$ and $(1 - G^a(t))$ are denoted respectively by $\mathrm{FRR}_a(t)$ and $\mathrm{FAR}_a(t)$. In the following, we list several popular FR fairness metrics. All of them are used by the U.S. National Institute of Standards and Technology (NIST) in their FRVT report [Grother, 2022]. Those fairness metrics attempt to quantify the differentials in $(\mathrm{FAR}_a(t))_{a \in \mathcal{A}}$ and $(\mathrm{FRR}_a(t))_{a \in \mathcal{A}}$. Since each metric fairness has two versions (one for the differentials in terms of FAR, the other in terms of FRR), we only present its FAR version. All metrics depend here on the threshold $t_\alpha$ which satisfies $\mathrm{FAR}_{\mathrm{total}}(t_\alpha) = \alpha \in [0, 1]$, meaning that the threshold is set so that it achieves a FAR equal to $\alpha$ for the global population, and not for some specific subgroup.

**Max-min ratio.** This metric has also been introduced by Conti et al. [2022], but for another choice of threshold $t_\alpha$. Its advantage is to be very interpretable but it is sensitive to low values in the denominator.

$$\mathrm{FAR}_{\mathrm{min}}^{\mathrm{max}}(\alpha) = \frac{\max_{a \in \mathcal{A}} \mathrm{FAR}_a(t_\alpha)}{\min_{a \in \mathcal{A}} \mathrm{FAR}_a(t_\alpha)}.$$

**Max-geomean ratio.** This metric replaces the previous minimum by the geometric mean $\mathrm{FAR}^\dagger(t_\alpha)$ of the values $(\mathrm{FAR}_a(t_\alpha))_{a \in \mathcal{A}}$, in order to be less sensitive to low values in the denominator.

$$\mathrm{FAR}_{\mathrm{geomean}}^{\mathrm{max}}(\alpha) = \frac{\max_{a \in \mathcal{A}} \mathrm{FAR}_a(t_\alpha)}{\mathrm{FAR}^\dagger(t_\alpha)}.$$

**Log-geomean sum.** It is a sum of normalized logarithms.

$$\mathrm{FAR}_{\mathrm{geomean}}^{\mathrm{log}}(\alpha) = \sum_{a \in \mathcal{A}} \left| \log_{10} \frac{\mathrm{FAR}_a(t_\alpha)}{\mathrm{FAR}^\dagger(t_\alpha)} \right|.$$

**Gini coefficient.** The Gini coefficient is a measure of inequality in a population. It ranges from a minimum value of zero, when all individuals are equal, to a theoretical maximum of one in an infinite population in which every individual except one has a size of zero.

$$\mathrm{FAR}_{\mathrm{Gini}}(\alpha) = \frac{|\mathcal{A}|}{|\mathcal{A}| - 1} \frac{\sum_{a \in \mathcal{A}} \sum_{b \in \mathcal{A}} |\mathrm{FAR}_a(t_\alpha) - \mathrm{FAR}_b(t_\alpha)|}{2 |\mathcal{A}|^2 \mathrm{FAR}^\dagger(t_\alpha)}.$$

Conti et al. [2022] argue that the choice of a threshold $t_\alpha$ achieving a global $\mathrm{FAR}_{\mathrm{total}} = \alpha$ is not entirely relevant since it depends on the relative proportions of each sensitive attribute value $a$ within the evaluation dataset together with the relative proportion of intra-group impostors. They propose instead a threshold $t_\alpha$ such that each group $a$ satisfies $\mathrm{FAR}_a(t_\alpha) \leq \alpha$. Since we are dealing with a unique evaluation dataset, we do not use such a threshold choice, to be consistent with the last three fairness metrics. Other fairness metrics exist in the literature such as the maximum difference in the values $(\mathrm{FAR}_a(t_\alpha))_{a \in \mathcal{A}}$ used by Alasadi et al. [2019], Dhar et al. [2021]. They have the disadvantage of not being normalized and are thus not interpretable, especially when comparing their values at different levels $\alpha$.

## 2 Assessing the Uncertainty of Face Recognition Metrics through Bootstrap

As previously explained, the ROC curves (and their related scalar summaries) of a similarity scoring function $s(x, x')$ (determined in practice by an encoder function to which cosine similarity is next applied) provide the main tool to assess performance and fairness in face recognition. We now investigate how to bootstrap these functional criteria, in order to evaluate the uncertainty/variability inherent in their estimation based on (supposedly i.i.d.) sampling observations drawn from the statistical populations under study. Indeed, this evaluation is crucial to judge whether the similarity scoring function candidate meets the performance/fairness requirements in a trustworthy manner, as will be next discussed on real examples in the next section.

**Bootstrapping the ROC curve of a similarity scoring function.** Extending the limit results in Hiesh and Turnbull [1996], the consistency of the empirical ROC curve (1) of a similarity scoring function $s(x, x')$ can be classically established, as well as its asymptotic Gaussianity (under additional hypotheses, involving the absolute continuity of distributions $F$ and $G$ in particular), in a standard

multi-sample asymptotic framework, *i.e.* stipulating that, for all $k \in \{1, \ldots, K\}$, $n_k/N \to \lambda_k > 0$ as $N \to +\infty$). Indeed, under appropriate mild technical assumptions, one may prove that the sequence of stochastic processes

$$\left\{ \sqrt{N}(\mathrm{ROC}_N(\alpha) - \mathrm{ROC}(\alpha)) \right\}_{\alpha \in (0,1)}$$

converges in distribution to a Gaussian law as $N \to \infty$. However, this limit law can hardly be used to build (asymptotic) confidence bands for the true ROC curve (or confidence intervals for scalar summary ROC-based metrics) in practice, due to its great complexity (the limit law, depending on the unknown densities of $F$ and $G$ is built from Brownian bridges and its approximate numerical simulation is a considerable challenge). Resampling techniques must be used instead, in order to mimic the random fluctuations of $\mathrm{ROC}_N(\alpha) - \mathrm{ROC}(\alpha)$. Application of the (smoothed) bootstrap methodology to ROC analysis has been investigated at length in the bipartite ranking context, *i.e.* for binary classification data [Bertail et al., 2008]. In the classification framework, bootstrap versions of the empirical ROC curve are simply obtained by means of uniform sampling with replacement in the two statistical populations (positive and negative). In this case, the empirical true/false positive rates are of the form of i.i.d. averages, which greatly differs from the present situation, where $F_N(t)$ is an average of independent mono-sample $U$-statistics of degree 2, while $G_N(t)$ is a multi-sample $U$-statistic of degree $(1, 1)$. As will be shown below and illustrated in Appendix B, the pairwise nature of the statistical quantity $F_N(t)$ computed is of great consequence, insofar as a 'naive' implementation of the bootstrap completely fails to reproduce $\mathrm{ROC}_N$'s variability when applied to the latter. Indeed, it systematically leads to a serious underestimation of $F_N(t)$, and consequently to an underestimation of $\mathrm{ROC}_N$ uniformly on $(0, 1)$. For simplicity's sake, we describe the reason behind this phenomenon by considering the problem of bootstrapping the statistic $F_N(t)$ and explain next how to remedy this problem.

**Bootstrap of $F_N(t)$.** For all $1 \le k \le K$, consider $(X_{1*}^k, \ldots, X_{n_k*}^k)$ a bootstrap sample related to identity $k$, drawn by simple sampling with replacement from original data $\{(X_1^k, \ldots, X_{n_k}^k)\}$. Recall that the original statistic is of the form:

$$F_N(t) = \frac{1}{K} \sum_{k=1}^{K} F_N^k(t) \quad \text{with} \quad F_N^k(t) = \frac{1}{\binom{n_k}{2}} \sum_{1 \le i < j \le n_k} \mathbf{1}_{s(X_i^k, X_j^k) \le t}.$$

Using the previous bootstrap sample, we can compute a bootstrap version of $F_N^k(t)$:

$$F_{N*}^k(t) = \frac{1}{\binom{n_k}{2}} \sum_{1 \le i < j \le n_k} \mathbf{1}_{s(X_{i*}^k, X_{j*}^k) \le t}.$$

$F_N^k(t)$ is a (non degenerate) $U$-statistic of order 2 (an average over all pairs) with symmetric kernel $\mathbf{1}_{s(x, x') \le t}$, and thus involves no 'diagonal' terms of type $\mathbf{1}_{s(X_i^k, X_i^k) \le t}$. Indeed, evaluating the similarity of an image and itself brings no information (it is naturally equal to 1 when considering cosine similarity). By contrast, it is shown in Janssen [1997] that the bootstrap version $F_{N*}^k(t)$ of $F_N^k(t)$ is in expectation equal to its $V$-statistic version, *i.e.* the version obtained by incorporating the diagonal terms in the average. In details, denoting $\mathbf{E}^*[\cdot | X_1^k, \ldots, X_{n_k}^k]$ the conditional expectation with respect to $(X_1^k, \ldots, X_{n_k}^k)$ (i.e. it denotes the expectation related to the randomness induced by the resampling), we have that:

$$\mathbf{E}^*[F_{N*}^k(t) | X_1^k, \ldots, X_{n_k}^k] = \frac{1}{n_k^2} \sum_{1 \le i,j \le n_k} \mathbf{1}_{s(X_i^k, X_j^k) \le t}.$$

Grouping all $K$ identities, we can compute a bootstrap version of $F_N(t)$:

$$F_{N*}(t) = \frac{1}{K} \sum_{k=1}^{K} \frac{1}{\binom{n_k}{2}} \sum_{1 \le i < j \le n_k} \mathbf{1}_{s(X_{i*}^k, X_{j*}^k) \le t}$$

whose expectation is:

$$\overline{F}_{N*}(t) := \mathbf{E}^*[F_{N*}(t) | X_1^1, \ldots, X_{n_K}^K] = \frac{1}{K} \sum_{k=1}^{K} \frac{1}{n_k^2} \sum_{1 \le i,j \le n_k} \mathbf{1}_{s(X_i^k, X_j^k) \le t}.$$

This means that bootstrapping $F_N(t)$ would result in many values $F_{N*}(t)$ that are not centered around $F_N(t)$, but centered around $\overline{F}_{N*}(t)$. From Janssen [1997] (Theorem 3), we find that $\mathbf{P}[\sqrt{N}(F_{N*}(t) - \overline{F}_{N*}(t)) \leq x | X_1^1, \ldots, X_{n_K}^K]$ is a uniformly consistent estimator for $\mathbf{P}[\sqrt{N}(F_N(t) - F(t)) \leq x]$. As a consequence, we can build confidence intervals for $F_N(t)$ in the following way: from the bootstrap samples, build confidence interval for $(F_{N*}(t) - \overline{F}_{N*}(t))$ and shift it by $F_N(t)$.

**Bootstrap of $G_N^{-1}(\alpha)$.** By contrast, a naive bootstrap method, involving no recentering, can be applied to:

$$G_N(t) = \frac{1}{\binom{K}{2}} \sum_{1 \leq k < l \leq K} G_N^{k,l}(t) \quad \text{with} \quad G_N^{k,l}(t) = \frac{1}{n_k n_l} \sum_{\substack{1 \leq i \leq n_k \\ 1 \leq j \leq n_l}} \mathbf{1}_{s(X_i^k, X_j^l) \leq t}.$$

Since each sample of the 2-sample U-statistic $G_N^{k,l}(t)$ is of order $(1,1)$, we have:

$$\mathbf{E}^*[G_{N*}^{k,l}(t) | X_1^k, \ldots, X_{n_l}^l] = G_N^{k,l}(t) \quad \text{and} \quad \overline{G}_{N*}(t) = G_N(t).$$

The previous confidence interval method still works here: build a confidence interval for $(G_{N*}(t) - \overline{G}_{N*}(t)) = (G_{N*}(t) - G_N(t))$ and shift it by $G_N(t)$. However, in this case, the bootstrap values $G_{N*}(t)$ are centered around $G_N(t)$. This method for confidence interval construction extends naturally to the bootstrap of the empirical quantile function $G_N^{-1}$. In theory, a smoothed version of the bootstrap, consisting in sampling from smoothed (by means of a *e.g.* Gaussian kernel) versions of the empirical distributions should be used for bootstrapping quantiles. However, given the very large size of the pooled dataset here, smoothing can be ignored in practice.

**Bootstrap of $ROC_N(\alpha)$.** Finally, we regroup the bootstrap of $F_N(t)$ and of $G_N^{-1}(\alpha)$ to present the bootstrap of the empirical ROC curve $ROC_N(\alpha) = F_N \circ G_N^{-1}(1 - \alpha)$. Using many bootstrap samples, a confidence interval is found for $(ROC_{N*}(\alpha) - \overline{ROC}_{N*}(\alpha)) = (F_{N*} \circ G_{N*}^{-1}(1 - \alpha) - \overline{F}_{N*} \circ G_N^{-1}(1 - \alpha))$, then shifted by $ROC_N(\alpha)$. A pseudo-code for building the confidence interval for $ROC_N(\alpha)$ at level $\alpha_{CI} \in [0, 1]$ is summarized in Algorithm 1. We highlight the significance of the recentering step and why a naive bootstrap does not work in Appendix B.

**Bootstrapping fairness metrics** We apply the same bootstrap algorithm for all fairness metrics since they are functions of $F_N$ and $G_N$. In details, consider a fairness measure that depends on $FRR_a(t)$, the max-min ratio $FRR_{\min}^{\max}$ for instance. In the expression of $FRR_{\min}^{\max}$, $FRR_a(t)$ would be computed as $F_N(t)$ for the classic fairness measure (equivalent of $ROC_N$ above), as $F_{N*}(t)$ for the bootstrap fairness (equivalent of $ROC_{N*}$) and as $\overline{F}_{N*}(t)$ for the V-statistic fairness (equivalent of $\overline{ROC}_{N*}$), using pairs of attribute $a$ in all three cases. Using those values for each $a \in \mathcal{A}$, we take the maximum and divide it by the minimum to get three versions of the fairness metric $FRR_{\min}^{\max}$: the classic fairness (where each $FRR_a(t)$ is computed as $F_N(t)$), the bootstrap fairness (where each $FRR_a(t)$ is computed as $F_{N*}(t)$) and the V-statistic fairness (where each $FRR_a(t)$ is computed as $\overline{F}_{N*}(t)$). Finally, the same bootstrap algorithm than for $ROC_N$ is used: compute bootstrap samples of the difference between the bootstrap fairness and the V-statistic fairness, then shift it by the classic fairness measure. Confidence bounds can be obtained in the same way than for $ROC_N$.

## 3 Numerical Experiments - Discussion

We use as encoder the trained[1] model ArcFace [Deng et al., 2019a] whose CNN architecture is a ResNet100 [Han et al., 2017]. It has been trained on the MS1M-RetinaFace dataset, introduced by [Deng et al., 2019b] in the ICCV 2019 Lightweight Face Recognition Challenge. We choose the dataset RFW [Wang et al., 2019] as evaluation dataset. It is composed of 40k face images from 11k distinct identities. This dataset is also provided with ground-truth race labels (the four available labels are: African, Asian, Caucasian, Indian) and is widely used for fairness evaluation since it is equally distributed among the race subgroups, in terms of images and identities. The official RFW

---

[1] https://github.com/deepinsight/insightface/tree/master/recognition/arcface_torch.

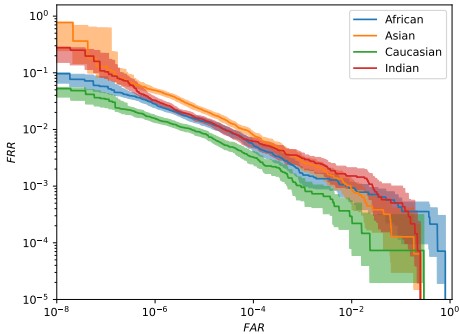

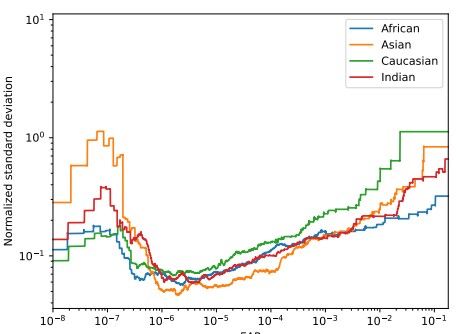

Figure 1: Confidence bands at 95% confidence level for the ROC of each race label. $B = 100$ bootstrap samples are used. The classic intra-group ROC curves are depicted as solid lines.

Figure 2: Normalized standard deviation of $B = 100$ intra-group bootstrap ROC curves, for each race label. The renormalization factor is the classic intra-group ROC curve.

protocol only considers a few matching pairs among all the possible pairs given the whole RFW dataset. The number of images is typically not enough to get good estimates of our fairness metrics at low FAR. To overcome this, we consider all possible same-race matching pairs among the whole RFW dataset. All images are pre-processed by the Retina-Face detector Deng et al. [2019c] and are of size $112 \times 112$ pixels.

Our first experiment is the computation of the confidence bands at 95% confidence level ($\alpha_{CI} = 0.05$) for each intra-group ROC i.e. the ROC corresponding to each race label. This is the output of our Algorithm 1 using $B = 100$ bootstrap samples and the result is displayed in Figure 1. It can be observed that Caucasians have a better performance than other races and that the uncertainty makes all races potentially indistinguishable in terms of performance at high FAR levels. Notice that the uncertainty increases when any of the error rates FAR, FRR is low, which happens when a few matching pairs are incorrectly classified, making the error rates really sensitive to those pairs. To quantify better the uncertainty in the estimation of $\mathrm{ROC}_N(\alpha)$, we compute the standard deviation of the $B = 100$ bootstrap ROC curves $\mathrm{ROC}_{N*}(\alpha)$, for each race label. For a fair comparison, we normalize this standard deviation by $\mathrm{ROC}_N(\alpha)$ (classic evaluation). The result is provided as a function of $\alpha$ in Figure 2. This normalized standard deviation is a natural proxy measure for the uncertainty in the estimation of the ROC of each race label. It is worth noting that the higher uncertainty is achieved by Asians and Indians at low FAR levels and by Caucasians at high FAR levels. Note that Caucasians have the best performance at low FAR levels and, at the same time, the lowest uncertainty about it among all race labels.

Then, we investigate the uncertainty related to certain possible fairness measures. The race label is used here as the sensitive attribute $a$. We compute the previous normalized standard deviation for the considered fairness metrics, in the same way than for Figure 2. For each metric, we take $B = 100$ bootstrap samples, giving 100 fairness values at each $\mathrm{FAR}_{\text{total}} = \alpha$ level. For each $\alpha$, the standard deviation of those values is found, and then normalized by the classic fairness measure at this level $\alpha$, for a fair comparison. As illustrated in Figure 3, the Gini coefficient and the log-geomean sum fairness metrics show high (similar) uncertainty. The max-geomean ratio metric displays the lowest uncertainty, both in terms of FAR and FRR, which makes it particularly suitable for fairness evaluation. In addition, the max-geomean (and the max-min) ratio metrics have the significant advantage to be interpretable.

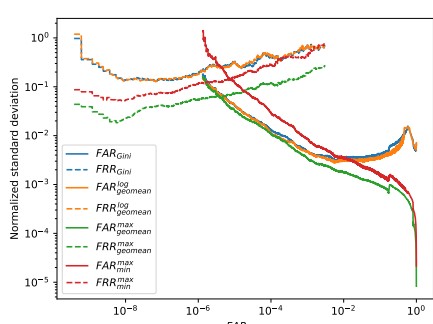

Figure 3: Normalized standard deviation of $B = 100$ bootstrap fairness curves (as functions of $\mathrm{FAR}_{\text{total}}$), for each fairness metric. The renormalization factor is the classic fairness measure.

While the gold standard by which fairness will be evaluated in the future is not fixed yet, we believe that it should definitely incorporate uncertainty measures, since it could lead to wrong conclusions otherwise. The bootstrap approach is simple, fast and yet it has not been explored by the FR community.

## Acknowledgments and Disclosure of Funding

This research was partially supported by the French National Research Agency (ANR), under grant ANR-20-CE23-0028 (LIMPID project).

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

# A    Pseudo-code for the bootstrap of $\mathrm{ROC}_N(\alpha)$

---

**Algorithm 1** Bootstrap of $\mathrm{ROC}_N(\alpha)$

---

**Input:** $K \geq 0$, images $(X_1^1, \ldots, X_{n_K}^K)$, encoder $f$
**Require:** $\alpha \in [0, 1]$, $B \geq 0$, $\alpha_{CI} \in [0, 1]$
**Output:** $\mathrm{CI}_-, \mathrm{CI}_+$, bounds for the confidence interval of $\mathrm{ROC}_N(\alpha)$ at level $\alpha_{CI}$
$\quad \overline{\mathrm{ROC}}_{N*} \leftarrow \overline{F}_{N*} \circ G_N^{-1}(1 - \alpha)$
$\quad \mathrm{gap} \leftarrow \emptyset$
$\quad$ **for** $b \leftarrow 1, B$ **do**
$\quad\quad X_{(b)} \leftarrow \emptyset$
$\quad\quad$ **for** $k \leftarrow 1, K$ **do**
$\quad\quad\quad X_{(b)}^k \leftarrow$ sample with replacement $n_k$ images among $(X_1^k, \ldots, X_{n_k}^k)$
$\quad\quad\quad X_{(b)} \leftarrow X_{(b)} \cup X_{(b)}^k$
$\quad\quad$ **end for**
$\quad\quad \mathrm{ROC}_{N,(b)} \leftarrow F_{N*} \circ G_{N*}^{-1}(1 - \alpha)$ for bootstrap sample $X_{(b)}$
$\quad\quad \mathrm{gap}_{(b)} \leftarrow \mathrm{ROC}_{N,(b)} - \overline{\mathrm{ROC}}_{N*}$
$\quad\quad \mathrm{gap} \leftarrow \mathrm{gap} \cup \mathrm{gap}_{(b)}$
$\quad$ **end for**
$\quad \mathrm{CI}_- \leftarrow \dfrac{\alpha_{CI}}{2}$-th quantile of gap
$\quad \mathrm{CI}_+ \leftarrow (1 - \dfrac{\alpha_{CI}}{2})$-th quantile of gap
$\quad \mathrm{ROC}_N \leftarrow F_N \circ G_N^{-1}(1 - \alpha)$
$\quad \mathrm{CI}_- \leftarrow \mathrm{ROC}_N + \mathrm{CI}_-$
$\quad \mathrm{CI}_+ \leftarrow \mathrm{ROC}_N + \mathrm{CI}_+$

---

# B    Visualization of the recentering step

In this section, we underline the significance of the recentering step of Algorithm 1. For the sake of simplicity, we achieve the bootstrap of the ROC curve for the global population, and not for some specific subgroups.

Let suppose that a naive bootstrap is done, that is we get some bootstrap image samples and, for each of them, we compute the bootstrap version $\mathrm{ROC}_{N*}$ of $\mathrm{ROC}_N$. If a naive bootstrap is achieved, the bootstrap versions $\mathrm{ROC}_{N*}$ (for many bootstrap samples) would be supposed to be centered around $\mathrm{ROC}_N$. By taking quantiles of $\mathrm{ROC}_{N*}(\alpha)$ for a given FAR level equal to $\alpha$, we would get the confidence interval at this FAR level $\alpha$. However, as illustrated in Figure 4 and Figure 5, this is not the case. The theoretical reasons have been detailed in Section 2. Briefly, since $(\mathrm{ROC}_{N*}(\alpha) - \overline{\mathrm{ROC}}_{N*}(\alpha))$ is a good estimator of $(\mathrm{ROC}_N(\alpha) - \mathrm{ROC}(\alpha))$, we can obtain confidence intervals for the latter with confidence intervals for the former.

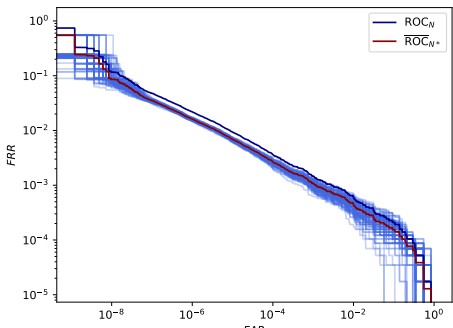

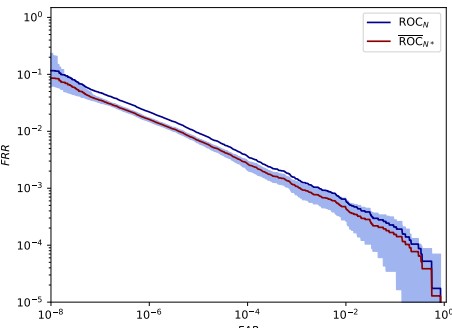

Figure 4: Bootstrap versions $\mathrm{ROC}_{N*}$ of the ROC curve for the global population of the RFW dataset. $B = 100$ bootstrap samples are considered. The classic version $\mathrm{ROC}_N$ is depicted as a dark-blue solid line while its V-statistic version $\overline{\mathrm{ROC}}_{N*}$ is depicted as a red solid line.

Figure 5: Confidence bands at 95% confidence level for the bootstrap versions $\mathrm{ROC}_{N*}$ of the ROC curve for the global population of the RFW dataset. $B = 100$ bootstrap samples are considered. The classic version $\mathrm{ROC}_N$ is depicted as a dark-blue solid line while its V-statistic version $\overline{\mathrm{ROC}}_{N*}$ is depicted as a red solid line.

