# OpenReview forum: "Assessing Performance and Fairness Metrics in Face Recognition - Bootstrap Methods"
_NeurIPS.cc/2022/Workshop/TSRML — TSRML2022_

### Official Review · Reviewer_J2ow · 2022-10-11
**Interesting analysis of bootstrapping for ROC curves & fairness for face recognition problems**

**Overall Rating:** 7

**Summary:**

The authors consider the aim of gaining insights into ROC curves obtained for face recognition tasks when fairness is desired.
In particular, they wish to provide interesting insights by generating confidence intervals thanks to bootstrapping.

The authors describe how using naive bootstrapping to produce confidence intervals in the considered setting has a systematic bias that needs to be accounted for.
They propose a simple algorithm to do this that involves recentering.
They then show the effect of this on commonly-used fairness metrics.

**Strengths:**

- The problem setting is, for the most part, clear and IMHO of interest to the venue
- The necessary background is given
- The proposed improvement (fixing naive bootstrapping to obtain CIs) is explained well and IMHO of interest to the venue
- The paper is well written


**Weaknesses:**

- A little unclear what the connection is between getting empirical C.I. for ROC curves and considering fairness metrics.
- A little unclear why specifically *face* recognition, it seems to me that your contribution can be generalized to other recognition problems.

- Line 51 and definitions of $F$ and $G$ are confusing and, I believe, incorrect. Firstly, it looks like by "negative" you mean $y_1 = y_2$, i.e., two images represent the same person. I guess this is an anomaly detection setting: maybe good to state this? Else, I think one would normally assume that "positive" means recognizing two images of the same person as such.
Secondly, you say that $F$ is False Negative but then $s$ is smaller than $t$ while $y_1=y_2$. I think it should be $y_1 \neq y_2$; in other words, you should swap the definition of $F$ with that of $G$ (or, in the text of line 51, firstly mention True Negative and then False Negative).

Minor points:
- Line 21: you say that the objective of FR is learning embeddings, but IMHO the objective is to recognize faces. Learning embeddings is the way in which many SotA methods (e.g., deep nets) happen to work nowadays.
- Line 22 "same identities" -> I guess you mean "some identities"
- Lines 62-66: Why not use FRR and FAR directly from the beginning?



**Overall Recommendation:**

This is the first time I see a paper on bootstrapping for obtaining CIs for ROC curves (irrespective of fairness considerations) on recognition problems. To the best of my understanding, this is a key difference between the presented work and Bertail et al. 2008. I thus assume the author's contribution (recentering to account for the recognition case) is novel.

Given the premise above, It seems to me this is a fair and interesting contribution. Software packages should consider implementing this method.
It remains a bit unclear to me why specifically focus on *face* recognition as a motivation for this. It seems to me this is just linked to the specific experiment carried out.

**Review Confidence:**

3: The reviewer is fairly confident that the evaluation is correct

---

### Official Review · Reviewer_ySLJ · 2022-10-19

**Overall Rating:** 6

**Summary:**

The paper investigates the bootstrap approach for evaluating uncertainty of the ROC curve and associated fairness metrics in face verification task. The paper claims that the evaluation pipeline in face verification involves sampling positive and negative pairs of images, which makes it inappropriate to use simple bootstrapping for ROC curve because of underestimation of false rejection rate. Instead, the authors propose to re-center the bootstrap confidence intervals. Finally, the authors use their method to estimate uncertainty of face verification model performance across different races in RFW dataset as well as uncertainty of different fairness statistics.

**Strengths:**

1. The problem of estimating uncertainty in face recognition models and their fairness is important.
2. The authors approach the problem from theoretical perspective and justify the need of re-centering the bootstrap confidence interval for false rejection rate and the associated metrics (ROC curve, fairness metrics.). They provide an algorithm for computing re-centered bootstrap confidence intervals.
3. The authors conduct experiments on popular face verification dataset and analyse uncertainty of ROC curve across races as well as uncertainty of certain fairness statistics.

**Weaknesses:**

Section 2 with theoretical justification for re-centering the bootstrap confidence intervals is very hard to read. It contains two pages of text with formulas, which are not organized in theorems/lemmas or even subsections. I suggest re-organizing chapter 2 to make it easier to follow.

**Overall Recommendation:**

Although, I believe that some of the parts of the paper could be re-organized for better readability, I recommend this paper for acceptance because of its contribution to improving uncertainty estimation in face verification.

**Review Confidence:**

3: The reviewer is fairly confident that the evaluation is correct

---

### Official Review · Reviewer_gHfo · 2022-10-21
**Review of Paper**

**Overall Rating:** 5

**Summary:**

The paper proposes a ROC-based approach for evaluating Face Recognition methods in terms of performance and fairness. Furthermore, to determine the uncertainty of the existing metrics, the authors apply the bootstrap technique and find appropriate confidence intervals.

**Strengths:**

- The idea of evaluating fairness in fairness recognition tasks and having confidence intervals for them is quite interesting.

- The experiments are diverse, and the author's interpretation gives insight into why considering the confidence intervals alongside the point estimation of measures is crucial in many applications.


**Weaknesses:**

- The fairness measure is limited to demographic parity, and no other notions of fairness are discussed.

- The connection between fairness and uncertainty sets is not clear. The bootstrap technique can be independently used for estimating the performance measure (FAR and FRR). The authors have failed to show why using the presented techniques is useful when we measure the fairness of FR tasks.


**Overall Recommendation:**

The paper requires more clarification on the relation between the proposed bootstrap technique and fairness measures. Otherwise, the main contribution of the paper remains unclear.

**Review Confidence:**

3: The reviewer is fairly confident that the evaluation is correct

---

### Decision · Program_Chairs · 2022-10-23

**Decision:**

Accept

**Comment:**

This paper studies the fairness vs performance tradeoff in facial recognition, with a specific focus on bootstrapping methods. The contents are relevant to this workshop and the results are novel. I hope the authors can improve the clarity of the paper based on reviews.